# DISTFLOW: A Fully Distributed RL Framework for Scalable and Efficient LLM Post-Training

Zhixin Wang [1 2]  Jiaming Xu [2 4]  Tianyi Zhou [2 3]  Mingjun Zhang [5]  Liming Liu [2]  Jiarui Hu [2]  Dian Yang [2]
Tongyu Wang [2]  Ping Zhang [5]  Jinlong Hou [2]  Siyuan Feng [2]  Yuan Qi [2 3]  Yuan Cheng [2 3]

## Abstract

Effectively scaling Reinforcement Learning (RL) is crucial for enhancing the reasoning and alignment of Large Language Models. The massive data and complex execution flows inherent in these tasks require a distributed architecture capable of efficient scaling. However, to simplify programming and dependency management, mainstream frameworks often rely on a centralized architecture where a single node dispatches both control and data. This inherent coupling creates significant communication bottlenecks, severely limiting system scalability and efficiency. We present DISTFLOW, a novel, fully distributed RL framework that adopts a multi-controller paradigm. By decoupling data transmission from control dispatch, DISTFLOW establishes a parallelism-aware, decentralized Data Coordinator that leverages local caching, load balancing, and asynchronous double buffer to minimize communication overhead and mitigate straggler effects. For control logic, it introduces a task scheduler built upon Directed Acyclic Graph (DAG) that facilitates fine-grained, independent execution. Experimental results demonstrate that DISTFLOW achieves near-linear scalability up to 512 GPUs and delivers up to a 2.63x throughput improvement over state-of-the-art (SOTA) frameworks. The source code is available at: https://github.com/sii-research/siiRL.

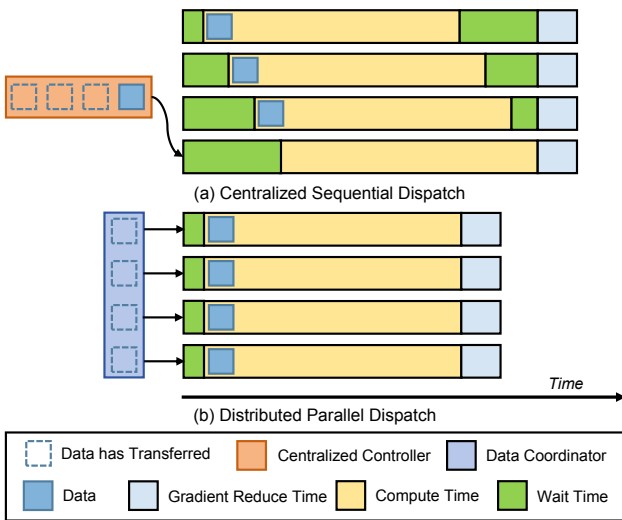

*Figure 1.* Comparison of data dispatch paradigms. (a) **Centralized Sequential Dispatch** creates accumulated idle time (green). (b) **Distributed Parallel Dispatch** enables concurrent transfer, eliminating serialization bottlenecks.

## 1. Introduction

The standard training paradigm for Large Language Models (LLMs) and Vision Language Models (VLMs) begins with pretraining (Vaswani et al., 2017) and Supervised Fine-Tuning (SFT) (Ouyang et al., 2022; Chung et al., 2024). Although SFT establishes basic instruction following, it often fails to guarantee robust alignment or complex reasoning. To address these limitations, modern pipelines incorporate Reinforcement Learning (Christiano et al., 2017) as a critical third stage. Essential for leading models like Deepseek-R1 (DeepSeek-AI, 2025), Claude 4.5 (Anthropic, 2025) and GPT-5 (OpenAI, 2025), RL shifts the training process from static data imitation to dynamic, goal-directed optimization.

Mainstream algorithms like Proximal Policy Optimization (PPO) (Schulman et al., 2017) and Group Relative Policy Optimization (GRPO) (Shao et al., 2024) rely on an iterative cycle comprising generation, evaluation, and training. While this workflow can be formally modeled as a DAG,

---

[1]Zhejiang University, Hangzhou, Zhejiang, China [2]Shanghai Innovation Institute, Shanghai, China [3]Fudan University, Shanghai, China [4]Shanghai Jiao Tong University, Shanghai, China [5]Infrawaves, Shanghai, China. Correspondence to: Siyuan Feng <syfeng@sii.edu.cn>, Yuan Qi <qiyuan@fudan.edu.cn>, Yuan Cheng <cheng_yuan@fudan.edu.cn>.

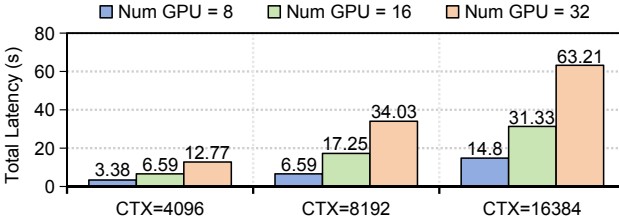

*Figure 2.* Latency profiling of the centralized controller in verl with a 7B LLM. " CTX " denotes context length.

the necessity of employing heterogeneous parallelization strategies across these distinct stages introduces significant complexity in coordinating data and control flows at scale.

Traditional RL systems, such as OpenRLHF (Hu et al., 2024), employed a disaggregated architecture, partitioning the system into distinct services for inference and training. This architecture enables flexible resource specialization through stage-specific optimization. However, its strict synchronization requirements forcing each stage to wait for the previous one to complete. This sequential execution results in significant resource idleness and low GPU utilization. Moreover, this separation introduces substantial data transfer overhead between the services. These limitations severely decrease the throughput of the system.

Researchers adopted colocated architectures to address the efficiency issues of disaggregated architectures. In this paradigm, the generation and training stages are executed on the same set of computational resources, with the system alternating between these two phases. This approach eliminates resource idleness and reduces data communication overhead. Building on this, frameworks like verl (Sheng et al., 2025b) have introduced hybrid controller paradigm that merges the flexibility of single-controller with the efficiency of multi-controller, thereby improving system throughput.

However, such architectures introduce new challenges. Although hybrid controller evenly dispatches the computation operation to multi-controller, the dataflow is managed by single-controller, including initial dataset loading and the collection and dispatch of vast intermediate data. The centralized mode forces all data to flow through a single node, creating significant I/O and communication overhead that becomes a severe bottleneck. As illustrated in Figure 1 (a), this centralized sequential dispatch forces workers to sit idle while waiting for data chunks, creating accumulated latency that delays the entire computation pipeline. Furthermore, quantitative analysis in Figure 2 reveals that this dispatch overhead scales linearly with both the cluster size (number of GPUs) and data volume (e.g., context length and batch size). This linear growth pattern renders the centralized architecture unsustainable for large-scale distributed systems. Consequently, when scaling the system to thousands

of GPUs, this centralized architecture is overwhelmed by the massive volume of data, leading to instability and crashes. As an illustrative profiling case, Appendix B shows that, for one GRPO step on 32 GPUs, verl spends substantial time in repeated split/dispatch/collect/concat operations, whereas DISTFLOW avoids redundant centralized data movement and limits communication to necessary data redistribution.

To address these limitations, we introduce DISTFLOW, a fully distributed RL training framework that achieves high throughput efficiency and near-linear scalability. DIST-FLOW fundamentally decouples control flow from data flow: control logic is assigned to the DAG Worker, while the data lifecycle is exclusively managed by the Data Coordinator. In this design, the control flow contains only lightweight execution metadata, such as DAG dependencies, worker assignment, parallelism configuration, task dispatch, and synchronization signals, whereas the data flow contains large RL payloads and data operations, such as prompts and responses, token-level tensors, log probabilities, rewards, advantages, and DataBuffer operations.

This separation keeps large intermediate tensors in decentralized data paths, while DAG Workers exchange only compact control information. Specifically, built on a DAG execution model, the DAG Worker utilizes a DAG Planner to translate logical algorithms into linearized task chains, effectively separating algorithmic logic from physical resources. Complementarily, the Data Coordinator eliminates centralized bottlenecks by decentralizing data management through distributed dataloaders and buffers, ensuring robust data flow across varying parallel strategies. Furthermore, by integrating local cache, load balancing, and asynchronous double buffer, DISTFLOW effectively minimizes communication overhead and straggler effects, enabling linear scalability up to 512 GPUs with remarkable runtime efficiency.

To validate the effectiveness of our framework, we conduct a comprehensive experimental evaluation. The results show that DISTFLOW exhibits exceptional performance and linear scalability across various cluster configurations, ranging from a single node to a 512 GPU scale. Compared to current SOTA colocated frameworks, DISTFLOW achieves up to a 2.63x speedup in end-to-end training throughput across different scenarios.

The main contributions of this work can be summarized as follows:

- We analyze the core performance bottlenecks of existing colocated RL frameworks, identifying the centralized controller as a critical constraint on both scalability and efficiency.
- We introduce DISTFLOW, a fully distributed framework built on a DAG execution model. DISTFLOW achieves a fundamental decoupling of control and data

flows: control logic is managed by DAG Workers, while the data lifecycle is governed by the Data Coordinator, with local caching, load balancing, and asynchronous double buffer integrated to minimize communication overhead and straggler effects.

- We conduct extensive evaluations of DISTFLOW against SOTA systems. Our results demonstrate near-linear scalability up to a 512 GPU scale and show significant end-to-end throughput improvements across various algorithms, model sizes, and model types, reaching up to 2.63x in specific scenarios.

## 2. Related Works

**Disaggregated Architectures.** The disaggregated architecture assigns different models (e.g., Actor, Critic) to dedicated GPU resources to isolate computation. Early frameworks, such as OpenRLHF (Hu et al., 2024) and NeMo-Aligner (Shen et al., 2024), adopted this design for its implementation simplicity and logical clarity. However, due to the strict serial dependencies between RLHF stages, this approach traditionally suffers from severe pipeline bubbles and resource under-utilization. To mitigate these inefficiencies, recent frameworks such as StreamRL (Zhong et al., 2025a), AReaL (Fu et al., 2026), RollPacker (Gao et al., 2026), and Laminar (Sheng et al., 2025a) have revisited this paradigm. By exploiting techniques such as stream generation and asynchronous pipelines, these modern systems aim to mask communication latency and unlock the potential of disaggregated resources in large-scale heterogeneous environments. However, such asynchronous execution strategies inevitably introduce data staleness and off-policy discrepancies, which can negatively impact the model's convergence performance and training stability.

**Colocated Architectures.** In contrast, the colocated architecture deploys all computation stages on the same set of GPUs using time-sharing scheduling, aiming to maximize memory and compute utilization. Pioneered by DeepSpeed-Chat (Yao et al., 2023), this approach eliminates the overhead of transferring data between separate model clusters. Subsequent research has significantly refined this paradigm to enhance resource efficiency. For instance, verl (Sheng et al., 2025b) optimizes the scheduling mechanism, while RLHFuse (Zhong et al., 2025b) introduces subtask fusion to further reduce pipeline bubbles. Additionally, other works have focused on optimizing the efficiency of model weight switching across different stages (Sheng et al., 2025b), solidifying the colocated architecture as a standard for resource-constrained training scenarios.

## 3. Motivation

### 3.1. Limitations of Existing RL Systems

**Centralized Controller.** Furthermore, this centralized architecture imposes a hard scalability ceiling. As the cluster expands, the linearly growing coordination overhead quickly saturates the controller node's fixed bandwidth. This bottleneck causes dispatch latency to exceed computation time, rendering the central node incapable of efficiently driving large-scale clusters.

**Coupling of Control and Data Management.** The single-controller paradigm suffers from a critical architectural defect: the coupling of command dispatch and data transport. By forcing the central node to orchestrate both execution and massive data movements, this design creates a dual bottleneck. The controller effectively becomes a serializer for parallel workloads, where handling vast intermediate tensors consumes the resources needed for scheduling. This dependency prevents the control plane from scaling efficiently, leading to inevitable system instability and crashes under high-load distributed scenarios.

### 3.2. Design Principles

RL workflows are uniquely characterized by frequent transitions between heterogeneous parallel strategies, which necessitate intricate tensor redistribution. While centralized orchestration simplifies logic management, it incurs prohibitive overhead when mediating massive data communications at scale. Conversely, a decentralized data plane is inherently advantageous for handling such complex flows due to its scalability and flexibility. With these insights, we propose a fully distributed architecture for DISTFLOW, like shown in Figure 1 (b). Our key design principle is to fundamentally decouple the control flow from the data flow. This design ensures that data flow is isolated from control flow, incorporating local caching, load balancing, and asynchronous double buffer as core mechanisms directly within the data plane. Ultimately, by combining this decentralized data governance with a DAG-defined execution model that separates algorithmic logic from physical resource management, DISTFLOW achieves high throughput and linear scalability.

## 4. DISTFLOW Overview

Based on the design principle of decoupling control and data flows, we propose DISTFLOW, a fully distributed RL framework engineered for large-scale scalability. As illustrated in Figure 3, the architecture separates the system into distinct operational components. The DAG Planner (§5.1) translates high-level logical graphs into serialized task chains, while DAG Workers (§5.2) orchestrate the control flow on each GPU, executing specific computational tasks without direct

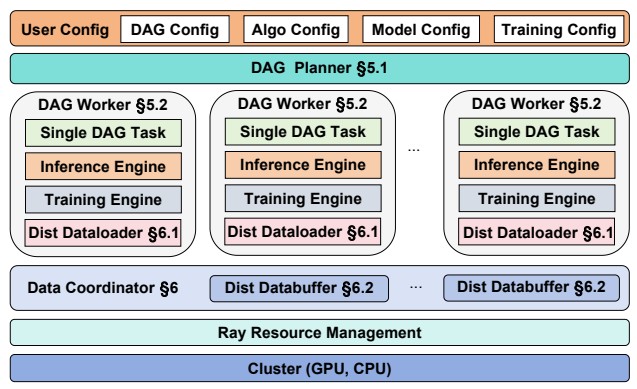

Figure 3. Overview of DISTFLOW.

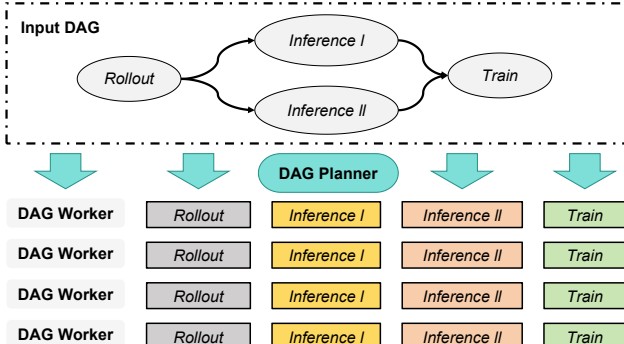

Figure 4. Decomposing a user-defined DAG into a sequential execution pipeline.

management of the underlying data transport.

Complementing this execution logic, the Data Coordinator (§6) autonomously governs the data flow. It manages the entire data lifecycle through distributed dataloading and buffering, while integrating local caching, load balancing, and asynchronous double buffer as core mechanisms. These components work in concert to handle the intricate tensor redistribution required by parallel strategy transitions, ensuring efficient data movement independent of the control logic.

We implemented our system based on PyTorch (Paszke et al., 2019). For resource management of GPU and CPU resources, we use Ray (Moritz et al., 2018) which is an open source framework to build and scale ML and Python applications easily. Our system's architecture integrates specialized engines for different stages. We use PyTorch Fully Sharded Data Parallel (FSDP) (Paszke et al., 2019) and Megatron (Shoeybi et al., 2020) as the training engine. For generation stage, we utilize the vLLM (Kwon et al., 2023) and SGLang (Zheng et al., 2024) inference engines. To manage these components, and inspired by the hierarchical API design of verl , our system uses the 3DParallelWorker base class.

## 5. DAG Workflow

To address the complexity of implementing diverse RL workflows, our framework is centered on a DAG-defined execution model. The core design principle is to decouple the logical representation of the algorithmic workflow from its physical computation resource. This separation of concerns is achieved through two main components: a declarative DAG interface for users and a backend DAG Planner that translates the logical graph into an executable task chain.

### 5.1. DAG Planner

**Input DAG Definition.** Users define the RL workflow via a declarative DAG configuration, where nodes represent

computational primitives characterized by Node ID, Role, Type, and Dependencies. This high-level abstraction enables the system to automatically derive the execution topology, shielding users from the complexities of distributed scheduling.

**DAG Decomposition.** A primary challenge in executing a user-defined DAG is ensuring its efficient adaptation to a colocated architecture with limited resources. Our framework addresses this challenge through the DAG Planner. Its fundamental responsibility is to translate the logical graph into a concrete, linearized execution pipeline that avoids resource contention and potential OOM errors. To achieve this, the planner automatically serializes the workflow by analyzing the logical depth of each node. If multiple nodes exist at the same depth, which would imply parallel execution, the planner systematically introduces dependencies to enforce a sequential order. For instance, as illustrated in Figure 4, if an input DAG contains two parallel nodes, Inference I and Inference II, the planner transforms the graph by making one a prerequisite for the other.

### 5.2. DAG Worker

To translate the logical workflow into execution, the DAG Worker operates on a single GPU through a structured lifecycle and dynamic function dispatch. The lifecycle begins with an Initialization phase, where the worker instantiates backend engines (e.g., vLLM, Megatron) and utilizes the dispatch mechanism to bind abstract nodes to concrete functions. This process materializes the serialized task chain into an executable queue, effectively decoupling algorithmic logic from system implementation.

During Execution, the worker iterates through the task chain, using a databuffer to pass outputs from one node as inputs to the next. This separation of structure and logic facilitates rapid innovation by allowing researchers to implement a new algorithm simply by defining a custom function and mapping it to the DAG.

**Algorithm 1** Constrained LPT Load Balancing

---

**Require:** Sequence lengths $\mathcal{S} = [s_0, \ldots, s_{N-1}]$, $K$ workers
**Ensure:** Balanced partitions $\mathcal{P}$
1: $I \leftarrow \text{argsort}(\mathcal{S}, \text{descending})$
2: $N \leftarrow |\mathcal{S}|$
3: **Assert** $N \bmod K = 0$
4: $C \leftarrow N/K$
5: $H \leftarrow \{(0, k) \mid k \in \{0, \ldots, K-1\}\}$
6: $\mathcal{P} \leftarrow \{\emptyset\}_{k=0}^{K-1}$
7: **for each** $u \in I$ **do**
8:      $B \leftarrow \emptyset$
9:      **while** True **do**
10:          $(load, k) \leftarrow H.\text{pop\_min}()$
11:          **if** $|P_k| < C$ **then**
12:              $P_k \leftarrow P_k \cup \{u\}$
13:              $H.\text{push}((load + \mathcal{S}[u], k))$
14:              **break**
15:          **else**
16:              $B \leftarrow B \cup \{(load, k)\}$
17:      **for each** $(l, k) \in B$ **do**
18:          $H.\text{push}((l, k))$
19: **return** $\mathcal{P}$

---

## 6. Data Coordinator

To address the scalability bottlenecks of centralized loading and the complexity of dynamic tensor redistribution, we introduce the Data Coordinator. As a unified abstraction governing the entire data lifecycle, it manages both static partitioning and transient data flow. By consolidating these responsibilities, the framework achieves a strict separation of data flow from control flow, providing a robust foundation for scalable execution.

### 6.1. Distributed Dataloader

In large-scale scenarios, a centralized approach where one node loads the entire dataset is fundamentally inefficient and unscalable. Therefore, to address this, our framework implements a Distributed Dataloader. The number of Distributed Dataloaders equals the number of DAG Workers, i.e., the number of GPUs. During initialization, the Distributed Dataloader queries the worker's parallelism strategy to partition the global dataset and exclusively loads the shard corresponding to its Data Parallelism (DP) rank, as illustrated in Figure 6. This approach inherently avoids single-node memory bottlenecks and achieves higher data loading efficiency through parallelism.

### 6.2. Distributed Databuffer

The Distributed Databuffer, a core component for data flow, is responsible for data redistribution between RL stages.

One instance is allocated per node and shared by local workers. Its primary function is to act as a parallelism-aware intermediary, ensuring both the correctness and efficiency of data flow during stage transitions where the DP sizes of consecutive stages may differ.

Upon stage completion, the Databuffer collects outputs exclusively from the DAG Worker with a Tensor Parallelism (TP) rank of 0 to prevent data redundancy. The subsequent operational path depends on the parallelism configuration of the next stage. When the DP size remains constant, the system executes a fast-path operation via a local cache mechanism. As illustrated in left of the Figure 5 by a transition from DP=2 to DP=4, this automated handling guarantees correct data flow and load balancing across varying parallel strategies.

### 6.3. Local Cache

To minimize communication overhead and serialization latency from Ray object management, we implement local caching in each DAG Worker. Since interacting with the Distributed Databuffer adds extra system load, the framework avoids unnecessary data movement whenever possible. When the DP size remains consistent between consecutive stages, intermediate data stays in the worker's local cache, bypassing the external Databuffer. Data is routed through the distributed Ray store only when mismatched parallelism strategies require re-partitioning, so costly remote operations occur only when topology transformation is actually necessary.

### 6.4. Load Balancing

To mitigate straggler effects caused by variable sequence lengths in training stage, the DataBuffer implements a Constrained Longest Processing Time (LPT) heuristic. Unlike standard greedy approaches, this algorithm balances the total computational load by prioritizing the assignment of long sequences to the least-loaded workers, while strictly enforcing a cardinality constraint. This guarantees that each worker receives an identical number of items (limit $C$), satisfying the synchronization requirements of collective communications while maximizing parallel efficiency.

### 6.5. Double Buffer

To eliminate pipeline stalls during epoch transitions, the DataBuffer employs an asynchronous Double Buffer Reset. Instead of synchronous clearing, the system performs an atomic pointer swap ($O(1)$) to instantly instantiate a fresh buffer. Physical memory deallocation is offloaded to a background task, thereby masking garbage collection overhead and ensuring continuous system throughput.

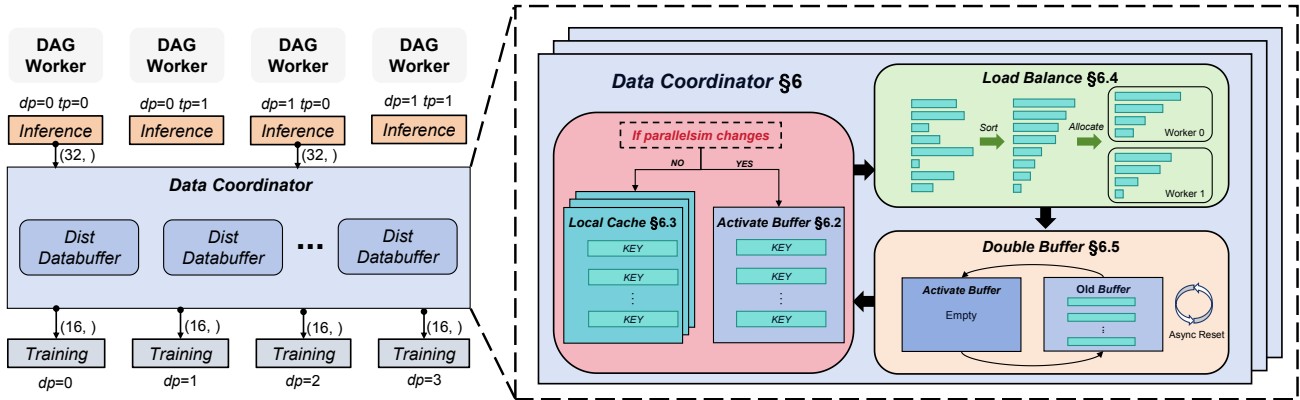

*Figure 5.* Workflow of the Data Coordinator. The left panel illustrates the data flow between DAG nodes (e.g., Inference to Training) across changing parallelism strategies. The right panel details the internal mechanisms, including strategy-aware local caching, dynamic load balancing, and double buffering to ensure efficient data transferring and minimal communication overhead.

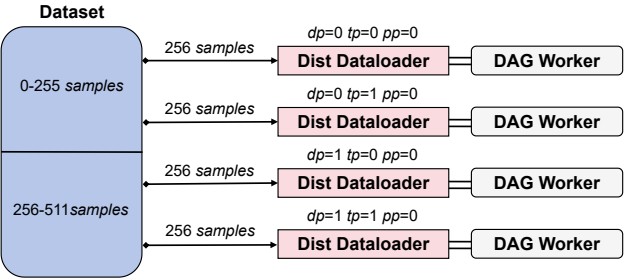

*Figure 6.* Workflow of Distributed Dataloader. Each worker is only responsible for loading its own assigned piece of the total data.

## 7. Evaluation

### 7.1. Experimental Setup

We conduct a series of experiments to evaluate DIST-FLOW's efficiency and scalability across four key scenarios. First, we assess overall performance on language models from 7B to 72B using PPO and GRPO algorithms, scaling up to 128 GPUs. Second, we test the linear scalability of DISTFLOW with VLMs on up to 512 GPUs using the GRPO algorithm; Third, we conduct ablation studies using the GRPO algorithm on 32 GPUs with a 7B LLM to analyze the impact of individual components in DISTFLOW. Finally, a convergence test is run for 20 epochs to ensure that DIST-FLOW's performance improvements do not compromise model accuracy. Additional long context experiments are detailed in the Appendix.

**Testbed.** We deploy DISTFLOW on a cluster with 64 nodes, where each node is equipped with 8 NVIDIA Hopper GPUs interconnected with NVLink. The nodes are connected by an RDMA network over RoCE v2. Our evaluation is conducted under the software settings with PyTorch 2.6.0, CUDA 12.6, vLLM 0.8.5.post1, and NCCL 2.21.5.

**Models and Algorithms.** We evaluate system performance using the PPO and GRPO algorithms. For the PPO experiments, a function reward is utilized in place of a reward model, with the critic model matching the actor's size. We use the Qwen-2.5-Instruct series for language models and the Qwen-2.5-VL-Instruct series for VLMs, with model sizes of 7B, 32B, and 72B.

**Datasets.** For language model setting, we use DeepScaleR-Preview-Dataset (Luo et al., 2025), which contains about 40,000 unique math problems, while for VLM experiments, we choose MM-Eureka-Dataset (Meng et al., 2025). All experiments are under the default maximum prompt length 2048, and the maximum response length 4096, with padding applied to shorter responses.

**Baseline.** We benchmark DISTFLOW against verl (Sheng et al., 2025b), a SOTA RL training system. Other frameworks (Hu et al., 2024; Rasley et al., 2020; Shen et al., 2024) are not selected for comparison due to their lower throughput relative to verl. Notably, we exclude asynchronous frameworks (e.g., StreamRL, AReaL) from this comparison. These systems typically achieve higher throughput by relaxing synchronization constraints, which inevitably compromises model convergence and algorithmic correctness. Both DISTFLOW and verl use vLLM as an inference engine and PyTorch FSDP as the training backend. The primary performance metric is throughput, measured in tokens per second, and is calculated from the total tokens in a global batch divided by the time for one iteration. Results are averaged over several iterations following a warm-up period to ensure accuracy.

### 7.2. End-to-End Evaluation

In our end-to-end evaluation, shown in Figure 7 and Figure 8, DISTFLOW consistently outperforms the baseline across all tested configurations. Our framework's advantage

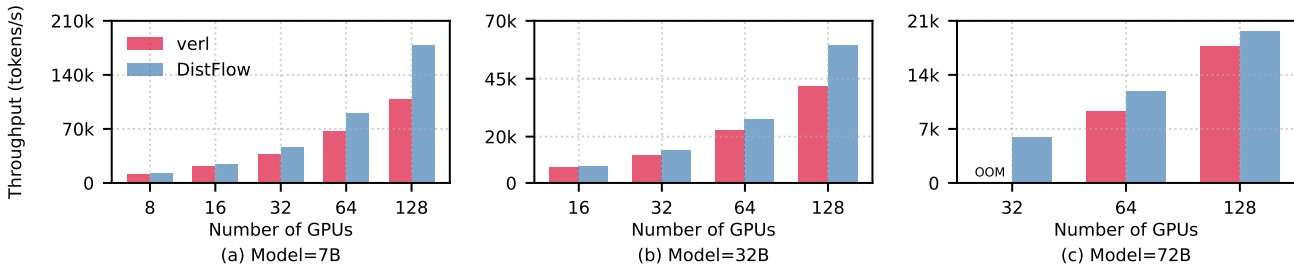

*Figure 7.* Throughput comparison of DISTFLOW and verl using the PPO algorithm. The results show that DISTFLOW is faster than the baseline for all tested model sizes and GPU counts. This speedup increases as more GPUs are added, and DISTFLOW can successfully complete large-scale tasks.

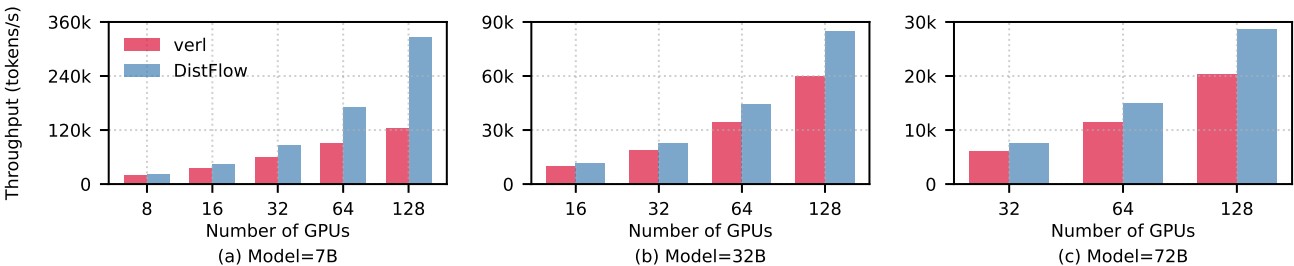

*Figure 8.* Throughput comparison of DISTFLOW and verl using the GRPO algorithm. With this more data-intensive algorithm, DISTFLOW's speed advantage becomes even greater, as its distributed data system handles the increased data load more efficiently.

is most pronounced in data-intensive scenarios. For PPO algorithm (Figure 7), we achieve 1.09x - 1.64x speedup comparing to the baseline. Remarkably, with the GRPO algorithm (Figure 8), which involves a larger data volume, DISTFLOW achieves a speedup of up to 2.62x. This highlights how our architecture excels where the baseline's centralized data handling fails.

This performance gap widens as computational resources increase. As we scale to more GPUs, the baseline's single-node bottleneck becomes more severe, lowering its throughput. In contrast, DISTFLOW's performance scales effectively, with its speedup over the baseline increasing with the GPU count. The baseline's architectural limits are made clear when it produces an OOM error with the 72B model on 32 GPUs, a task DISTFLOW handles without issue. Additionally, smaller models, which have a higher communication-to-compute ratio, benefit most. By optimizing the dataflow that constitutes a larger portion of their runtime, DISTFLOW delivers a 2.26x speedup for the 7B model on 128 GPUs, demonstrating the profound efficiency gains of our distributed approach.

### 7.3. Scalability Evaluation

The practical benefits of DISTFLOW's architecture are clearly demonstrated in our scalability experiments, conducted using the GRPO algorithm with VLMs on the MM-Eureka-Dataset. In this experiment, we scale the global batch size proportionally with the number of nodes. As

shown in Figure 9, the resulting performance (solid line) closely tracks the ideal linear scalability curve (dotted line). We quantify this linearity using the Scaling Efficiency metric, defined as follows:

$$\text{Scaling Efficiency} = \frac{T_2/T_1}{N_2/N_1} \times 100\% \qquad (1)$$

where $T$ is throughput and $N$ is the number of GPUs, with $(N_1, T_1)$ representing the baseline and $(N_2, T_2)$ the scaled configuration. The evaluation reveals excellent linearity across all model sizes. Specifically, DISTFLOW achieves remarkable scaling efficiencies of 90.1%, 93.9%, and 91.8% for the 7B, 32B, and 72B models, respectively, when scaling across hundreds of GPUs.

The experimental results from our scalability evaluation directly highlight the benefits of our framework's fully distributed design. Figure 9 demonstrates near-linear scaling, a critical capability for efficient large-scale training from 32 GPUs up to 512 GPUs. Such consistent performance indicates that the system effectively distributes all workloads, including data communication, thus avoiding the bottlenecks that typically hinder performance as a cluster grows. In contrast, the baseline system could not complete the same linearity tests, encountering OOM errors.

### 7.4. Ablation Study

To quantify the contribution of each module in DISTFLOW, we performed a stepwise ablation study across batch sizes

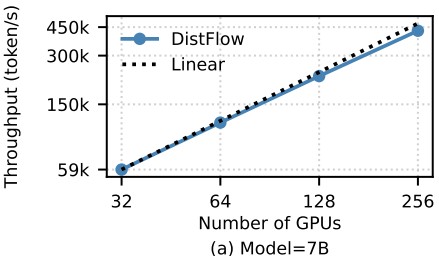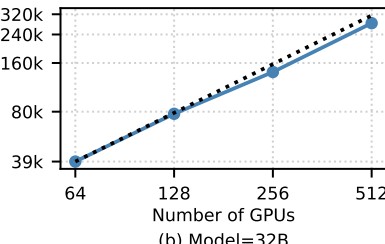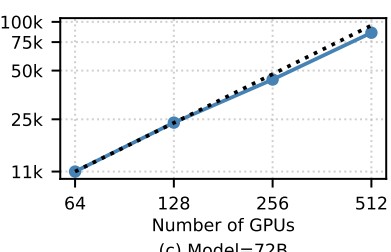

*Figure 9.* Scalability evaluation of DISTFLOW. This experiment shows that DISTFLOW achieves near linear scalability on large clusters of up to 512 GPUs. This strong performance is attributed to its fully distributed architecture, which uniformly balances both computational and dataflow workloads to maintain high efficiency at scale.

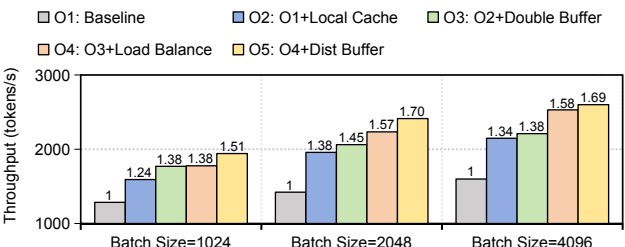

*Figure 10.* Ablation study of the system with different batch sizes.

of 1024, 2048, and 4096. As shown in Figure 10, starting from the Baseline (O1), we incrementally integrated Local Cache (O2), Double Buffering (O3), Load Balancing (O4), and Distributed Data Buffer (O5). We focus specifically on the DataCoordinator's internal components, excluding the Distributed DataLoader as it primarily targets startup latency and memory rather than throughput.

**Local Cache.** The integration of local cache yielded the most significant immediate performance gain across all batch sizes (e.g., a 1.38x speedup at Batch Size=2048). This confirms that redundant data retrieval and I/O overhead constituted the primary bottleneck in the baseline system. By caching pre-processed data shards, the system effectively eliminates repetitive storage access.

**Double Buffer.** The addition of the double buffer mechanism provided a consistent throughput improvement (increasing speedup from 1.38x to 1.45x at Batch Size=2048). This validates that the asynchronous reset strategy successfully overlaps data preparation with GPU computation, effectively masking the latency overhead associated with memory reclamation and Python garbage collection during iteration transitions.

**Load Balancing.** The effectiveness of load balancing is strongly tied to the batch size. For a small batch size (Batch Size=1024), the benefit was minimal as scheduling costs offset the gains. In contrast, the benefit was substantial at larger batch sizes. Specifically, at Batch Size=4096, the speedup rose from 1.38x to 1.58x. This confirms that our Constrained LPT algorithm effectively handles the straggler

effect, which becomes more pronounced as the number of sequences and the likelihood of length variance increase.

**Distributed Buffer.** Finally, the distributed buffer architecture provided a further performance uplift across all settings, culminating in a peak speedup of 1.70x (Batch Size=2048) and 1.69x (Batch Size=4096). To isolate the benefits of distribution, we simulated a centralized baseline by restricting the number of DataBuffer to 1 within the 4 node cluster, contrasting it with the fully distributed configuration. This comparison indicates that optimizing distributed architecture and coordination significantly reduces contention and communication overhead in the data pipeline.

### 7.5. Convergence

To verify that performance gains do not compromise accuracy, we compared DISTFLOW against verl by training a 32B model using GRPO on 32 GPUs (DeepScaleR-Preview-Dataset, 20 epochs). As shown in Figure 11, under identical hyperparameters, the reward and entropy curves of DIST-FLOW closely match the baseline. DISTFLOW achieves this parity while reducing total execution time by 21%, confirming that its efficiency gains come at no cost to accuracy.

## 8. Conclusion

This paper introduces DISTFLOW, a fully distributed framework designed to eliminate the scalability bottlenecks of centralized RL training. This decoupling allocates execution logic to DAG Workers and data governance to the Data Coordinator. The system assigns execution logic to DAG Workers via a linearized DAG model. In parallel, the Data Coordinator governs the data flow by managing the entire data lifecycle with local caching, load balancing, and asynchronous double buffer to minimize communication overhead and mitigate straggler effects. Extensive evaluations demonstrate that DISTFLOW achieves near linear scalability up to 512 GPUs and delivers up to a 2.63x improvement in end-to-end throughput compared to SOTA colocated frameworks. We believe DISTFLOW paves the

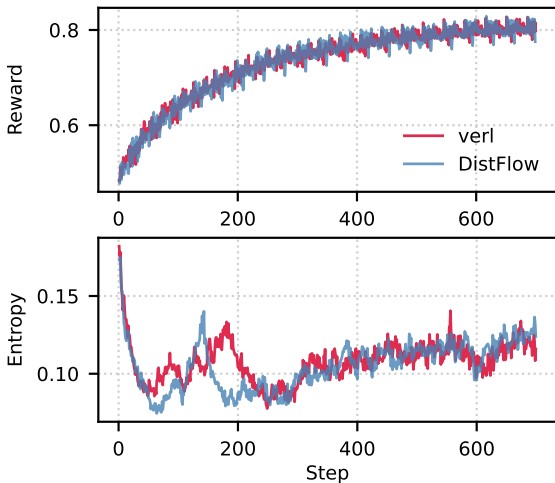

*Figure 11.* Entropy (up) and Reward (down) curves comparison between verl and DISTFLOW.

way for next-generation RL by enabling the training of frontier models with unprecedented scale and efficiency.

## Impact Statement

This paper presents work whose goal is to advance the field of Machine Learning. There are many potential societal consequences of our work, none which we feel must be specifically highlighted here.

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

# A. Additional Experiments

## A.1. Long-Context Evaluation

Long-context capability is a critical frontier for LLMs, particularly for the development of advanced agent systems that must process extensive histories or documents. These scenarios create highly data-intensive workloads where the sheer volume of token data can overwhelm a system's communication fabric. Our evaluation (Figure 12) in these long-context settings demonstrates that DISTFLOW's fully distributed dataflow provides a significant and scalable advantage. By allowing each node to manage its portion of the data, our framework avoids the severe communication overhead that troubles centralized systems, where all data must be funneled through a single, congested point.

The results, presented in Figure 12, confirm this advantage empirically. For the 7B model, DISTFLOW's throughput speedup over the baseline progressively grows from 1.48x at an 8k context length to an impressive 2.03x at 64k. This clear trend is highly significant; it shows that as the data volume and complexity of the task increase with longer contexts, the efficiency gains from our distributed design become even more pronounced. This directly implies that for future, more demanding applications, DISTFLOW's architectural superiority is an even greater asset.

Furthermore, the baseline system encounters a critical OOM error with the 72B model at a 32k context length, a demanding task that DISTFLOW handles without issue. This is not merely a performance dip but a fundamental breakdown, which underscores the scalability limitations of a centralized data management approach. This failure highlights a practical ceiling on the complexity that such systems can manage. In contrast, DISTFLOW's ability to complete the task demonstrates its superior robustness and its capacity to push the boundaries of what is possible in data-intensive, long-context scenarios.

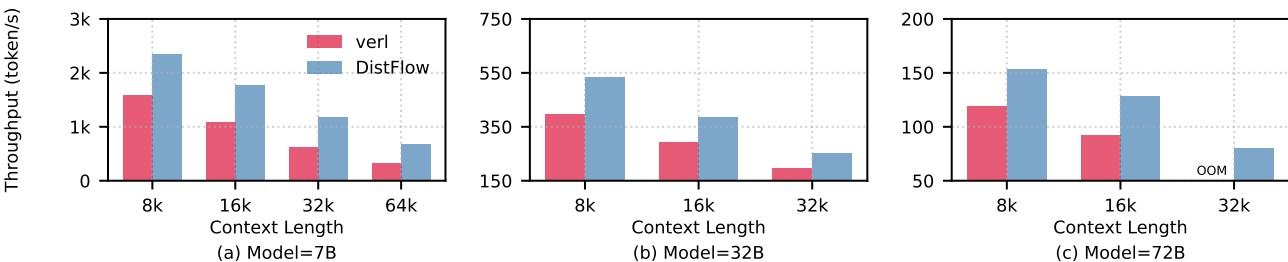

*Figure 12.* Long-context performance evaluation. The results show that DISTFLOW's performance advantage over the baseline increases with longer context lengths.

# B. Step Timeline Breakdown

*Table 1.* Procedure-level timeline of one GRPO step on GSM8K with 32 GPUs, per-node batch size 512, input/output length 8192, and 8 responses per prompt. All numbers are in seconds.

| | DISTFLOW | | | | verl | | | | | | |
|---|---|---|---|---|---|---|---|---|---|---|---|
| | Compute | Put/Get | Other | Total | Split | Dispatch | Compute | Collect | Concat | Other | Total |
| rollout | 29.236 | – | – | 29.236 | 0.003 | 0.709 | 25.169 | 9.825 | 6.392 | – | 42.098 |
| reward | 0.253 | – | – | 0.253 | – | – | 2.581 | – | – | – | 2.581 |
| advantage | 2.569 | 2.185 | – | 4.754 | – | – | 1.318 | – | – | – | 1.318 |
| actor_old_log_prob | 10.068 | 8.220 | – | 18.288 | 0.002 | 25.154 | 7.665 | 0.975 | 0.636 | – | 34.432 |
| ref_log_prob | 6.505 | – | – | 6.505 | 0.002 | 25.306 | 7.251 | 0.303 | 0.298 | – | 33.160 |
| actor_update | 32.565 | – | – | 32.565 | 0.002 | 32.044 | 35.902 | 0.007 | 0.001 | – | 67.956 |
| other overhead | – | – | 0.280 | 0.280 | – | – | – | – | – | 10.781 | 10.781 |
| total | 81.196 | 10.405 | 0.280 | 91.881 | 0.009 | 83.213 | 79.886 | 11.110 | 7.327 | 10.781 | 192.326 |

Table 1 separates backend compute from data handling. In verl, centralized dispatch dominates the non-compute cost, especially for actor log-probability, reference log-probability, and actor update. In DISTFLOW, most transitions are local-cache hits; only the advantage-to-actor-log-probability transition changes DP layout and therefore triggers a put/get through the DataBuffer. The "other" column includes data loading, metrics, graph/runtime overhead, and synchronization outside the displayed stage methods.

