# OpenReview forum: "DistFlow: A Fully Distributed RL Framework for Scalable and Efficient LLM Post-Training"
_ICML.cc/2026/Conference — ICML 2026 regular_

### Official Review · Reviewer_ims6 · 2026-03-10

**Soundness:** 3
**Presentation:** 2
**Significance:** 3
**Originality:** 3
**Overall Recommendation:** 5
**Confidence:** 3

**Summary:**

This paper proposes **DistFlow**, a fully distributed reinforcement learning framework for large-scale LLM post-training, and argues that the main bottleneck lies in the existing framework like verl is the centralized management of data flow and control flow. To address this issue, the authors propose to decouple control flow from data flow. A DAG-based planner and worker system manages execution logic, while a separate Data Coordinator handles the data lifecycle through distributed dataloading, buffering, local caching, load balancing, and asynchronous double buffering. Empirically, DistFlow achieves 1.09×–1.64× speedup for PPO and up to 2.62×/2.63× for GRPO, while also scaling nearly linearly to 512 GPUs with over 90% scaling efficiency on several model sizes. These are all compared with verl without harming convergence and performance.

**Compliance With Llm Reviewing Policy:**

Affirmed.

**Final Justification:**

The author provides supplementary results on the comparisons with asynchronous baseline AReal, demonstrating a better trade-off between throughput and policy improvement. This makes the experiment part comprehensive. I choose to raise the score, given the clear motivation and comprehensive evaluations.

**Key Questions For Authors:**

1. Can you add on other frameworks' results, like Areal?
2. Could you please provide more detailed performance/convergence comparisons?

**Limitations:**

yes

**Strengths And Weaknesses:**

## Strengths
**Clear motivation and reasonable design choices**: The paper tackles an important and timely systems problem in LLM post-training: the scalability bottleneck caused by centralized data orchestration in existing RL frameworks. Its main idea—explicitly decoupling control flow from data flow—is conceptually clean and well motivated, and the proposed architecture is supported by several practical components, including distributed dataloading, distributed buffering, local caching, load balancing, and asynchronous double buffering, which together form a coherent end-to-end system design.

**Comprehensive evaluations**: The empirical evaluation is also a strong point: the authors test DISTFLOW across PPO and GRPO, multiple model scales from 7B to 72B, and VLM settings up to 512 GPUs, showing consistent throughput gains over verl, especially in more data-intensive settings where the advantage reaches up to 2.62x/2.63x, along with near-linear scaling at larger cluster sizes.

## Weaknesses:
**Lack of framework baselines**: Although DistFlow is presented as a general framework for large-scale RL post-training, the comparison is conducted primarily against verl and excludes other frameworks such as StreamRL and AReaL, which are arguably highly relevant baselines for the throughput/scalability discussion.

**Lack of comprehensive convergence evaluation**: The convergence analysis is also limited: it is based on a single 32B GRPO setting over 20 epochs, and the paper mainly shows reward/entropy parity rather than broader evidence across tasks, algorithms, or longer training horizons.

---

> ### Author Rebuttal · Authors · 2026-03-30
>
> # Response to Reviewer ims6
>
> We thank the reviewer for the comments. We will incorporate these suggestions in the revision. Below we address two concerns: broader baselines and convergence evidence.
>
> ## Q1. Can you add other framework results, such as AReaL?
>
> **A1.** We agree that the baseline set should be broader. In this rebuttal, besides `verl`, we add a sync baseline (`slime`) and an async baseline (`slime_async`). Table 1 summarizes the new throughput/runtime comparison.
>
> For fairness, all methods in Table 1 use the same setting. Under this setting, DistFlow is fastest in all tested 8/16/32-GPU cases.
>
> We agree that frameworks such as AReaL are relevant. Due to limited rebuttal time and engineering effort, we were not able to complete a clean AReaL reproduction in this round. Still, the newly added sync/async baselines already broaden the comparison beyond `verl`, and we will further expand this part in the revision.
>
> ## Q2. Could you please provide more detailed performance/convergence comparisons?
>
> **A2.** We agree that the original convergence evidence was too limited. We therefore add 100-step comparisons on GSM8K-GRPO at 8/16/32 GPUs, GSM8K-PPO at 8 GPUs, and Geo3K-GRPO (multimodal) at 8 GPUs. To reduce per-step noise, all values are EMA-smoothed. `Reward@1` and `Reward@100` denote the smoothed `critic/rewards/mean` at step 1 and step 100.
>
> We also add three optimization indicators. `entropy_loss` reflects policy exploration/uncertainty, `grad_norm` reflects update stability, and `pg_clipfrac` is the fraction of samples affected by PPO/GRPO clipping. Since final-step `pg_clipfrac` can be too small to be informative, we report its 100-step average.
>
> Table 2 shows broader evidence across two algorithms, both text-only and multimodal tasks, and multiple GPU settings. Overall, DistFlow matches or exceeds `verl` in final reward while keeping comparable `grad_norm` and `pg_clipfrac`; `entropy_loss` may differ but shows no instability. This supports optimization consistency in addition to reward convergence, and we will add these results to the revision.
>
> ## Table 1. Supplementary baseline throughput/runtime comparison
>
> | Method | GPU | Per-GPU throughput | Total runtime |
> | --- | ---: | ---: | ---: |
> | slime | 8 | 1428.739 | 1738.596 |
> | slime | 16 | 583.644 | 2847.608 |
> | slime | 32 | 175.872 | 7959.207 |
> | slime_async | 8 | 1908.147 | 1360.387 |
> | slime_async | 16 | 1335.852 | 1978.381 |
> | slime_async | 32 | 698.717 | 3647.848 |
> | DistFlow | 8 | 2246.675 | 823.210 |
> | DistFlow | 16 | 2203.514 | 844.327 |
> | DistFlow | 32 | 1987.139 | 907.264 |
> | verl | 8 | 1746.841 | 1063.296 |
> | verl | 16 | 1418.372 | 1264.857 |
> | verl | 32 | 901.684 | 1975.789 |
>
> **Setup**: GRPO on GSM8K dataset, per-node batch size 512, global batch size proportional to node count, input/output length 8192, and 8 responses per prompt. DistFlow and verl use vLLM, while slime and slime_async use SGLang because slime only supports SGLang.
>
> ## Table 2. Supplementary convergence comparisons over 100 steps
>
> | Setting | GPU | Method | Reward@1 | Reward@100 | Entropy@100 | GradNorm@100 | Avg pg_clipfrac |
> | --- | ---: | --- | ---: | ---: | ---: | ---: | ---: |
> | GSM8K-GRPO | 8 | DistFlow | 0.7786 | 0.9762 | 0.0841 | 0.0410 | 8.1e-05 |
> |  |  | verl | 0.7637 | 0.9727 | 0.0562 | 0.0542 | 1.5e-04 |
> | GSM8K-GRPO | 16 | DistFlow | 0.7660 | 0.9784 | 0.0709 | 0.0295 | 1.2e-04 |
> |  |  | verl | 0.7600 | 0.9787 | 0.0599 | 0.0342 | 1.3e-04 |
> | GSM8K-GRPO | 32 | DistFlow | 0.7734 | 0.9828 | 0.0788 | 0.0195 | 1.2e-04 |
> |  |  | verl | 0.7725 | 0.9814 | 0.0724 | 0.0227 | 1.2e-04 |
> | GSM8K-PPO | 8 | DistFlow | 0.7695 | 0.9503 | 0.1875 | 0.6357 | 0.0e+00 |
> |  |  | verl | 0.7422 | 0.9524 | 0.0797 | 0.5314 | 0.0e+00 |
> | Geo3K-GRPO | 8 | DistFlow | 0.3199 | 0.6347 | 0.1480 | 0.5049 | 3.0e-03 |
> |  |  | verl | 0.3272 | 0.5713 | 0.1623 | 0.6490 | 3.2e-03 |

---

> > ### Author Rebuttal · Reviewer_ims6 · 2026-04-01
> >
> > The author didn't provide the throughput comparison to AReal as I requested, which I think could be the most direct comparison to Dist-Flow. Other concerns have been well resolved.

---

> > > ### Author Response · Authors · 2026-04-03
> > >
> > > # Second-Round Response to Reviewer ims6
> > >
> > > We thank the reviewer for the suggestion. We are sorry that, due to limited time and resources during the rebuttal period, we did not complete the AReaL experiment in time. We now provide this additional baseline under the same GSM8K-GRPO setting.
> > >
> > > ## Q1. Throughput comparison to AReal
> > >
> > > **A1.** Table 1 and Table 2 show three main observations.
> > >
> > > First, `AReaL` is consistently faster than `verl`, and is faster than `DistFlow` at `16/32` GPUs.
> > >
> > > Second, compared with the synchronous reference `DistFlow`, `AReaL` shows consistently lower reward, with reward gaps of `-0.0464/-0.0388/-0.0412` at `8/16/32` GPUs at 100 step.
> > >
> > > Overall, the added `AReaL` baseline broadens the comparison and supports the same conclusion: asynchronous designs can improve throughput, but `DistFlow` provides a stronger throughput-convergence tradeoff without sacrificing algorithmic quality. In addition, most DistFlow system optimizations, such as the distributed Databuffer, load balancing, and double buffer, are orthogonal to disaggregated async training and can also be applied there to further improve throughput.
> > >
> > > ## Table 1. Reward Comparison With AReaL
> > >
> > > | Method | GPU | Per-GPU throughput | Reward@100 | Reward gap vs DistFlow@step100 |
> > > | --- | ---: | ---: | ---: | ---: |
> > > | DistFlow | 8 | 2246.675 | 0.9762 | - |
> > > | AReaL | 8 | 2181.305 | 0.9298 | -0.0464 |
> > > | verl | 8 | 1746.841 | 0.9727 | - |
> > > | DistFlow | 16 | 2203.514 | 0.9784 | - |
> > > | AReaL | 16 | 2366.198 | 0.9396 | -0.0388 |
> > > | verl | 16 | 1418.372 | 0.9787 | - |
> > > | DistFlow | 32 | 1987.139 | 0.9828 | - |
> > > | AReaL | 32 | 2215.107 | 0.9416 | -0.0412 |
> > > | verl | 32 | 901.684 | 0.9814 | - |
> > >
> > > ## Table 2. Per-GPU Throughput by GPU Count
> > >
> > > | Method | 8 GPU | 16 GPU | 32 GPU |
> > > | --- | ---: | ---: | ---: |
> > > | DistFlow | 2246.675 | 2203.514 | 1987.139 |
> > > | AReaL | 2181.305 | 2366.198 | 2215.107 |
> > > | verl | 1746.841 | 1418.372 | 901.684 |
> > >
> > > **Setup**: `GSM8K-GRPO`, per-node batch size `512`, input/output length `8192`, and `8` responses per prompt. `AReaL` is run with the closest matching visible hyperparameters through its native `gsm8k_grpo` recipe. All runs use `100` steps.

---

### Official Review · Reviewer_g3K9 · 2026-03-13

**Soundness:** 3
**Presentation:** 2
**Significance:** 3
**Originality:** 2
**Overall Recommendation:** 4
**Confidence:** 5

**Summary:**

The paper argues that scaling reinforcement learning (RL) for large language models post-training requires a distributed system, as the workflow involves large data volumes and non-trivial execution dependencies across the generation, evaluation, and training stages. It argues that widely used RL post-training frameworks commonly rely on a centralised architecture in which a single node dispatches both control decisions and data transfers, and that this coupling creates communication and coordination bottlenecks that limit scalability and efficiency. To address this, the paper proposes DISTFLOW, a fully distributed RL framework with a multi-controller paradigm that separates control dispatch from data movement. It introduces (i) a decentralised Data Coordinator intended to manage the data lifecycle using mechanisms such as local caching, load balancing, and double buffering, and (ii) a DAG-based scheduling approach intended to represent RL workflows and execute them with more fine-grained task orchestration via DAG Workers.

**Compliance With Llm Reviewing Policy:**

Affirmed.

**Final Justification:**

The final updated decision is based on the additional experimental results.

**Key Questions For Authors:**

- Would you provide an explicit statement about what the data exchange paradigm is in the RL workflows implemented by DistFlow?

- Would you compare with more baselines, including both the synchronous and asynchronous systems?

**Limitations:**

Yes.

**Strengths And Weaknesses:**

Strengths:

- Distributed RL for LLMs is a timely and important system problem. The paper attempts to address a practically relevant bottleneck in large-scale RL post-training. The high-level design direction, i.e., decoupling control-flow scheduling from data-flow management, follows the general design principle of distributed systems.

Weaknesses:

- This paper is poorly written, i.e., it is very confusing about what exactly the so-called control plane and data plane are referring to for this RL workflow. Concretely, the paper repeatedly states that it decouples control from data by assigning control logic to DAG Workers and the data lifecycle to the Data Coordinator, but it does not provide a precise description for these planes (e.g., what messages belong to the control plane, and what objects belong to the data plane). The implementation discussion focuses on vague terms like “data buffers” but does not clearly enumerate the concrete objects in RL workflows, leaving it unclear what inference workers must exchange versus what should remain local.

- The evaluation explicitly excludes asynchronous frameworks and justifies the exclusion by asserting that relaxing synchronisation inevitably compromises convergence/correctness. However, recent asynchronous RL-for-LLMs systems explicitly target staleness control and report matched or even improved final performance compared to strong synchronous baselines while increasing training efficiency. Moreover, even the synchronous baseline (i.e., Verl) that the paper compares against already discusses asynchronous execution when models are placed on separate device pools, indicating that asynchrony is not inherently out of scope for the RL workflows (https://verl.readthedocs.io/en/latest/advance/fully_async.html).

- The paper benchmarks primarily against verl and argues that other frameworks are omitted because they are slower than verl, but this is not true given the existence of newer or more optimized **synchronous** baselines that substantially change the competitive landscape, including e.g., RollPacker (https://arxiv.org/abs/2509.21009).

---

> ### Author Rebuttal · Authors · 2026-03-30
>
> # Response to Reviewer g3K9
>
> We thank the reviewer for the comments. We will incorporate these points in the revision, clarify the control/data planes, revise the async and async-vs-sync discussion, add more sync and async baseline. Below we summarize these updates.
>
> ## Q1. Control And Data Plane?
>
> **A1.** In DistFlow, the **control plane** carries only lightweight execution metadata and ops: DAG structure, node dependencies, worker assignment, parallelism config, buffer keys, task dispatch, and ready/finish sync signals. It decides **who runs which stage and when**, but does not carry RL tensors. The **data plane** carries the RL batch payload and data ops: batch creation, local-cache store/load, DataBuffer put/get, repartition across DP changes, concat/split, and downstream batch use by reward, log-prob, reference, and update stages.
>
> Concretely, the data plane includes prompt/response tensors, input_ids, attention_mask, position_ids, response_mask, rollout log-probs, reward tensors, advantages, returns, old_log_probs, ref_log_prob, and sample metadata. More detailed timelines, tensor lists, and transfer sizes are in our responses to WkWT and vj9T.
>
> ## Q2. More Baselines?
>
> **A2.** Yes. We agree that the original baseline set was too narrow. In addition to `verl`, we now include a popular sync baseline (`slime`) and an async baseline (`slime_async`), where `slime_async` is a **one-step async** setting. Results are in Table 1 and Table 2. Following the reviewer's suggestion, we also tested `Roll(RollPacker)`, but it hit `OOM` under this config. The main reason is that ROLL's FSDP2 RLVR path materializes full-vocabulary FP32 logits during log-prob computation and does not use the packed/remove-padding training, which leads to a much higher peak memory footprint. These added baselines also support that the paper's conclusion is correct. We will add more baseline experiments in the camera-ready version if the paper is accepted.
>
> We also report scaling efficiency for all methods: `slime`/`slime_async` in Table 1 and `DistFlow`/`verl` in Table 2, using the paper's definition in section 7.3. These added baselines do not change the paper's main conclusion; they support that our design is effective, while making the comparison broader than using `verl` alone. Many of our system optimizations, including distributed DataBuffer, double buffering, and load balancing, are also compatible with decoupled async architectures and can be combined with async acceleration. We will revise the paper to avoid overstating `verl` as the full sync landscape. Due to limited rebuttal time and compute, we cannot add more baselines or larger-scale experiments in this round.
>
> ## Q3. Async And Convergence?
>
> **A3.** We agree that async should not be dismissed categorically. Our revised claim is narrower: under the same algorithm and config, async display reduced convergence quality because of stale samples, while async-aware algorithm changes can mitigate this issue. This is consistent with [AReaL](https://arxiv.org/abs/2505.24298), which also shows that async introduces a convergence loss. Algorithm changes can mitigate this effect, but not fully remove it. Our results show the same trend: in Table 1, `slime_async` improves efficiency over sync `slime`, but its reward is consistently lower. We will add a more detailed explanation and experimental discussion in the final paper.
>
> ## Table 1. Sync/async comparison
>
> | Method | GPU | per-GPU throughput | Scaling eff. | Async gain | Reward increase | Reward gap |
> | --- | ---: | ---: | ---: | ---: | ---: | ---: |
> | slime | 8 | 1428.739 | 100.00% | - | 0.176 | - |
> | slime_async | 8 | 1908.147 | 100.00% | +33.55% | 0.171 | -0.005 |
> | slime | 16 | 583.644 | 40.85% | - | 0.187 | - |
> | slime_async | 16 | 1335.852 | 70.01% | +128.88% | 0.166 | -0.021 |
> | slime | 32 | 175.872 | 12.31% | - | 0.178 | - |
> | slime_async | 32 | 698.717 | 36.62% | +297.29% | 0.167 | -0.011 |
>
> ## Table 2. Baseline throughput
>
> | Method | GPU | Per-GPU throughput | Scaling eff. | Total runtime |
> | --- | ---: | ---: | ---: | ---: |
> | DistFlow | 8 | 2246.675 | 100.00% | 823.210 |
> | DistFlow | 16 | 2203.514 | 98.08% | 844.327 |
> | DistFlow | 32 | 1987.139 | 88.45% | 907.264 |
> | verl | 8 | 1746.841 | 100.00% | 1063.296 |
> | verl | 16 | 1418.372 | 81.20% | 1264.857 |
> | verl | 32 | 901.684 | 51.62% | 1975.789 |
> | Roll (RollPacker) | - | OOM | - | - |
>
> **Setup** : GRPO on GSM8K dataset, per-node batch size 512, global batch size linear in node count, input/output length 8192, and 8 responses per prompt. DistFlow and verl use vLLM, while slime/slime_async use SGLang since slime supports only SGLang.

---

> > ### Author Rebuttal · Reviewer_g3K9 · 2026-04-01
> >
> > The author's reply resolved my concerns, and I raised my score accordingly.

---

> > > ### Author Response · Authors · 2026-04-07
> > >
> > > Thank you very much for your thoughtful review, detailed feedback, and follow-up acknowledgment. We sincerely appreciate your comments on both the presentation and the evaluation, which helped us significantly improve the clarity of the paper and broaden the discussion of baselines. We are grateful that our rebuttal addressed your concerns, and we will incorporate these improvements into the final revision. Thank you again for your time and support.

---

### Official Review · Reviewer_vj9T · 2026-03-13

**Soundness:** 3
**Presentation:** 3
**Significance:** 3
**Originality:** 3
**Overall Recommendation:** 5
**Confidence:** 3

**Summary:**

To tackle the overwhelming data flows in the single-controller architecture of distributed RL frameworks, especially when in a large scale, this paper decouples data and control flows by using multiple controllers. The evaluation with various scales of GPU-based systems has shown significant performance improvement over verl.

**Compliance With Llm Reviewing Policy:**

Affirmed.

**Final Justification:**

The authors have fully addressed my concerns, and I have raised my score.

**Key Questions For Authors:**

See weaknesses 1, 2, 3, 4.

**Limitations:**

Not applicable.

**Strengths And Weaknesses:**

Strength:

1. The paper is well motivated, well organized in structure, good in presentation and easy to follow.

2. The model sizes and experimental cluster scales selected in the paper are convincing.



Weakness:

1. This paper does not provide sufficient in-depth insights into the problem of the couple of control and data flows. For example, it does not explore factors influencing the level of the data transmission bottleneck in an RL framework.

2. The method design is more of an engineering effort. Though the data coordinator for decoupling data and control is an effective design, the design of its sub-components, including the data loader, buffer, and cache, is straightforward. It would be much better for the authors to highlight the novel (different from a common one or customized for the newly encountered problem) designs of these sub-components. It does not elaborate on the difficulties of simply refactoring Verl's single-controller into a pure multi-controller mode, which may introduce new problems.

3. Though the evaluation presents end-to-end throughput performance and an ablation study on different system components, it is unclear which procedure (rollout, inference, training, etc) during RL is improved. It is better to decompose the time into several procedures so that readers can accurately attribute the improved procedures and relate them to the effects of different system components. As data transmission improvement is supposed to be a significant effort of this paper, it is also advised to indicate the proportion of data transmission latency relative to overall RL computation.

4. Other questions:

a. How is the workload for collecting outputs from DAG workers in rank 0? How likely is rank 0 to become a new bottleneck?

b. How is the frequency of repartitioning? This is a concern as it affects both caching and load-balancing overhead.

c.  "Double Buffer" is used throughout the paper, whereas the title of Section 6.2 is "Distributed Databuffer." What is their relationship?

d.  In Figure 9, the x-axis increases uniformly while the y-axis does not (looks like it is increasing exponentially). What’s the rationale behind this for reporting "near-linear scalability"?

e. In Section 7.5, what's the possible source of the compromise of accuracy in DistFlow? Will DistFlow affect the consistency of computing results?

---

> ### Author Rebuttal · Authors · 2026-03-30
>
> # Response to Reviewer vj9T
>
> We thank the reviewer for the comments and will incorporate them in the revision.
>
> ## Q1. Data Bottleneck
>
> **A1.** In `verl`, the bottleneck comes from coupling data flow with control flow under a single controller. Control flow is light, but data flow is heavy. The controller dispatches payloads serially; if data are not ready, the control path cannot launch the next work. So the bottleneck is on data dispatch, and the control flow is blocked by it.
>
> `verl` time is mainly composed of data transfer (`dispatch/collect`), data handling (`split/concat`), and `compute`. `Compute` depends on the training and inference backends. Table 2 shows that the load of data handling and transfer grows almost linearly with cluster scale and data volume. This part contains redundant data movement and operations. DistFlow's Data Coordinator compresses this part aggressively: most data stay local, and data repartitioning is needed only when the DP layout changes. Therefore, as shown in Table 1, DistFlow spends 10.405 s on communication and data handling, while `verl` spends 101.66 s.
>
>
> ## Q2. System Design
>
> **A2.** RL workflows differ from traditional distributed systems because they carry large intermediate tensors and have more complex data dependencies. Data Coordinator should be viewed as a system, not isolated components. Local cache, distributed Databuffer, load balancing, double buffer, and future-based handoff minimize communication: data stay local when DP is unchanged, and cross-worker movement happens only on true DP changes, yielding zero redundant communication and speedups. The same design also manages data automatically and decouples it from control flow, improving flexibility for complex or disaggregated workflows. Also, verl's hybrid controller (single-controller + multi-controller) is a core design; refactoring it into multi-controller-only would require major architectural changes. Even then, without DistFlow's Data Coordinator, gains in performance and scalability would be limited.
>
> ## Q3. Other Questions
>
> **A3.**
>
> `a)` Rank 0 is unlikely to become a bottleneck. Over measured steps, mean rank0_collection_total_s = 0.010966 s, mean rank0_step_total_s = 33.808378 s, and the share is only 0.0325%. remote_get_calls = 0 and remote_put_calls = 0; extra work is mainly metrics_aggregation = 0.007190 s and reset_data_buffer = 0.003776 s.
>
> `b)` Repartitioning is triggered by DP changes, such as advantage -> actor_log_prob, not every stage. In the GRPO and PPO workflows, it happens only once.
>
> `c)` “Distributed Databuffer” stores intermediate tensors on DP-changed stages, while “Double Buffer” is one internal mechanism for overlapped switch/reset.
>
> `d)` The “near-linear scalability” claim is based on the scaling-efficiency definition in the paper. Since DistFlow achieves over 90% scaling efficiency, describing it as near-linear is reasonable. Because of the figure's plotting scale, the visual impression may differ.
>
> `e)` DistFlow's optimizations are lossless and do not sacrifice algorithmic effectiveness. Any small differences come from normal distributed-training nondeterminism, such as floating-point reduction order and batch sharding/order. See Table 2 in ims6.
>
>
> ## Table 1. Timeline
>
> | Stage | DistFlow breakdown | DistFlow total | verl breakdown | verl total |
> | --- | --- | ---: | --- | ---: |
> | rollout | compute 29.236s | 29.236 s | split 0.003s + dispatch 0.709s + compute 25.169s + collect 9.825s + concat 6.392s | 42.100 s |
> | reward | compute 0.253s | 0.253 s | compute 2.581s | 2.581 s |
> | advantage | compute 2.569s + PUT 2.185s | 4.754 s | compute 1.318s | 1.318 s |
> | actor_old_log_prob | GET 8.220s + compute 10.068s | 18.334 s | split 0.002s + dispatch 25.154s + compute 7.665s + collect 0.975s + concat 0.636s | 34.434 s |
> | ref_log_prob | compute 6.505s | 6.505 s | split 0.002s + dispatch 25.306s + compute 7.251s + collect 0.303s + concat 0.298s | 33.161 s |
> | actor_update | compute 32.565s | 32.565 s | split 0.002s + dispatch 32.044s + compute 35.902s + collect 0.007s + concat 0.001s | 67.959 s |
> | other overhead |  | 0.280 s |  | 10.781 s |
> | total |  | 91.926 s |  | 192.334 s |
>
> **Other step overhead**: denotes time outside the displayed stages/methods, including data loading, metrics, graph/runtime overhead, and other bookkeeping/synchronization.
>
> ## Table 2. verl Step-Level Time Breakdown
>
> | GPUs | nodes | Split (s) | Dispatch (s) | Compute (s) | Collect (s) | Concat (s) | Stage Time Sum (s) |
> | --- | --- | ---: | ---: | ---: | ---: | ---: | ---: |
> | 8 | 1 | 0.005 | 19.643 | 69.509 | 2.600 | 1.708 | 93.465 |
> | 16 | 2 | 0.004 | 36.810 | 76.719 | 4.993 | 2.760 | 121.285 |
> | 32 | 4 | 0.009 | 83.213 | 75.986 | 11.111 | 7.327 | 177.646 |
>
> **Setup** : Both exp use GRPO on GSM8K dataset, per-node batch size 512, global batch size linear in node count, input/output length 8192, and 8 responses per prompt.

---

> > ### Author Rebuttal · Reviewer_vj9T · 2026-04-01
> >
> > The authors have resolved all my concerns. I have raised my rating score accordingly.

---

> > > ### Author Response · Authors · 2026-04-07
> > >
> > > Thank you very much for your thoughtful review and constructive feedback. We sincerely appreciate your time and are grateful that our rebuttal addressed your concerns. Your comments have been very helpful, and we will incorporate the clarifications into the final revision. Thank you again for your support.

---

### Official Review · Reviewer_WkWT · 2026-03-21

**Soundness:** 3
**Presentation:** 3
**Significance:** 3
**Originality:** 3
**Overall Recommendation:** 4
**Confidence:** 4

**Summary:**

This paper proposes DistFlow, an optimized colocated framework for reinforcement learning that decouples the control flow (the definition of operations) from the data flow (the movement of inputs and outputs for each stage of the pipeline). Through a number of optimizations introduced by the authors, DistFlow achieves up to 2.63x higher training throughput compared to the baseline with no convergence differences.

**Compliance With Llm Reviewing Policy:**

Affirmed.

**Final Justification:**

The rebuttal addressed my main concerns, primarily reinforcing my prior assessment. I maintain my positive score for the submission

**Key Questions For Authors:**

1. Could you elaborate on the necessity of DAG-based formulation in DistFlow relative to all the optimizations you have introduced? From my understanding, it should be possible to have independent dispatch across workers and all the proposed improvements at the data layer without relying on the DAG abstraction. I would feel more confident in my rating if that point is explained clearly — potentially, with some additional clarification in the paper.
2. In Section 7.4, could you elaborate on the overheads associated with data movement, particularly those eliminated by local caching and the double buffer? It might be unclear to the reader why these optimizations are impactful, because the exact duration of the corresponding execution stages is not given. Having a summary of the data movement involved (in terms of tensor sizes and the communication bandwidth) would make it easier to both understand the motivation of the method and interpret the experiments.

**Limitations:**

Yes

**Strengths And Weaknesses:**

Strengths:
1. The proposed approach demonstrates strong gains across several reinforcement learning setups, and the ablation study quantifies the exact improvements from each component of DistFlow.
2. The topic of the paper is highly relevant to the current state of reinforcement learning frameworks, and the ideas proposed in the work are (to the best of my knowledge) quite novel.
3. DistFlow's implementation is available as supplementary material, which increases the reproducibility of the work.
4. Overall, the paper presents a comprehensive summary of the field in the related work section. The positioning of the method relative to prior and concurrent works is clear.

Weaknesses:
1. One minor aspect that could be improved is the display of data movement bottlenecks in standard colocated RL. Although the paper provides a clear explanation behind the optimizations, it is not fully clear why some of the inefficiencies (e.g., memory deallocation in between epoch transitions) have considerable impact on the training performance. Having a timeline view of a training iteration (for instance, a more detailed form of Figure 4) would be very helpful in explaining the advantage of proposed optimizations to the reader.
2. I might be mistaken, but it appears as if the effect of the decentralized controller is not fully quantified. Several optimizations proposed in Section 6 are applicable to the centralized setting, and as per Section 7.4, they correspond to a significant fraction of the overall speedup. However, it is not stated whether Figure 10 reports results for the PPO or the GRPO setting, as well as the model size chosen: therefore, it is unclear whether the results can be directly matched with Figure 2, which reports the overhead of the centralized controller. It would be quite useful to see a throughput comparison for centralized and decentralized dispatch, with all other optimizations kept the same across setups.

---

> ### Author Rebuttal · Authors · 2026-03-30
>
> # Response to Reviewer WkWT
>
> We thank the reviewer for the comments. We will incorporate these suggestions in the revision.
> ## Q1. Centralized Dispatch and DAG Design
>
> **A1.** We implemented a centralized-dispatch version and profiled it explicitly. The controller spends 0.0162 s/step on next-node scheduling and 0.0163 s/step on node-completion handling, for a combined cost of 0.0325 s/step, only 0.04% of the overall step time. The dominant costs are still graph execution and step synchronization, so the controller itself is not the bottleneck at this scale.
>
> The DAG is mainly for flexibility rather than performance. Different DAG workers can execute different DAGs, so DistFlow natively supports disaggregated architectures and future multi-agent joint training. It also separates workflow definition from physical worker layout. We will make this clearer in the revision with additional explanation in the paper.
>
> For Figure 10, the setting is already given in Section 7.1 and aligned with Figure 2.
>
> ## Q2. Timeline and Data Movement
>
> **A2.** We added detailed timeline and tensor profiling. The detailed timeline view is summarized in Table 1 and Table 2 of our response to reviewer vj9T, and the transferred tensors and sizes are summarized in Table 1 bottom here.
>
> First, the timeline makes the improved procedures explicit. In standard colocated RL with `verl`, one training step is mainly composed of `dispatch/collect`, `split/concat`, and `compute`. The detailed breakdown in Table 1 and Table 2 of our response to reviewer vj9T shows that the dominant extra cost is not backend compute itself, but the data-handling and data-transfer part around each RL procedure. In particular, `dispatch` and `collect` grow strongly with scale, while `compute` is relatively stable because it is determined by the training/inference backend. This is why the main gain of DistFlow is not from changing the backend compute, but from compressing the communication and data-handling path. The same timeline also shows that in DistFlow most stages stay local, and only the true DP-changing edge `advantage -> actor_old_log_prob` remains non-local.
>
> For the second point, the data movement overhead is large because in `verl` the controller must repeatedly dispatch and collect large intermediate tensors, and Ray also adds serialization/deserialization overhead on top of the raw transfer cost. This is why these stages can take so long in practice. The tensor summary in Table 1 bottom makes this concrete: in DistFlow, only `advantage -> actor_old_log_prob` triggers cross-worker movement, with 4865 MB in total and only two transfers (one put and one get). In contrast, one `verl` step performs eight transfers across the four major stages and moves about 43,107 MB in total. This explains why local cache is impactful: it removes repeated movement on same-DP stages and eliminates redundant communication. Double buffer is also impactful because it overlaps reset/deallocation with useful work, so reset is removed from the critical path and the measured overhead stays near zero.
>
>
> ## Table 1. RL-step tensors
>
> | Tensor | DistFlow stage | verl stage | Type | DistFlow MB | verl MB |
> | --- | --- | --- | --- | ---: | ---: |
> | prompts, responses | rollout, reward, advantage, actor_log_prob, ref_log_prob, actor_update | rollout, reward, actor_log_prob, ref_log_prob | int64 | 64 | 256 |
> | input_ids, attention_mask, position_ids | rollout, reward, advantage, actor_log_prob, ref_log_prob, actor_update | rollout, actor_log_prob, ref_log_prob | int64 | 128 | 512 |
> | response_mask | rollout, reward, advantage, actor_log_prob, actor_update | actor_log_prob | int64 | 64 | 256 |
> | rollout_log_probs | rollout, actor_update | rollout, actor_log_prob, ref_log_prob, actor_update | fp32 | 32 | 128 |
> | token_level_scores, token_level_rewards | reward, advantage, actor_update | reward, advantage, actor_update | fp32 | 32 | 128 |
> | advantages, returns | advantage, actor_log_prob, actor_update | advantage, actor_update | fp32 | 32 | 128 |
> | old_log_probs | actor_log_prob, ref_log_prob, actor_update | actor_log_prob, ref_log_prob, actor_update | fp32 | 32 | 128 |
> | entropys | actor_log_prob | actor_log_prob | fp32 | 32 | 128 |
> | ref_log_prob | ref_log_prob, actor_update | ref_log_prob, actor_update | fp32 | 32 | 128 |
>
> **Transfer**:
>
> `DistFlow`: only `advantage -> actor_log_prob` triggers cross-worker transfer, 4865 MB in total, with only two transfers.
>
> `verl`: one step performs eight transfers across four stages and moves about 43,107 MB in total.
>
> **Setup**: 4-node GRPO on GSM8K dataset, batch size 512/node, input/output length 8192, and 8 responses per prompt.

---

> > ### Author Rebuttal · Reviewer_WkWT · 2026-04-03
> >
> > I thank the authors for their response and detailed explanations. I have increased my confidence score and will maintain my overall evaluation of the paper.

---

> > > ### Author Response · Authors · 2026-04-07
> > >
> > > Thank you very much for your thoughtful review, detailed feedback, and follow-up acknowledgment. We sincerely appreciate your time and the constructive suggestions, which have helped us improve both the clarity and presentation of the paper. We are also grateful that our rebuttal addressed your concerns. Thank you again for your support and careful evaluation.

---

### Decision · Program_Chairs · 2026-04-30

**Decision:**

Accept (regular)

**Comment:**

### **Summary of Contributions**
The paper introduces DistFlow, a fully distributed reinforcement learning framework designed for the scalable post-training of Large Language Models (LLMs). Existing mainstream frameworks typically rely on a centralized architecture for dispatching both control and data, which creates significant communication bottlenecks as model sizes and context lengths scale. DistFlow resolves this by utilizing a decentralized data management approach, decoupling control signals from data transfer and employing asynchronous execution to maximize throughput and minimize wait times.

### **Decision Reasoning**
The submission received a strong consensus of positive scores from all four reviewers.

* **Strengths:** Reviewers commended the architectural shift from centralized to distributed data management, noting it as a timely and necessary evolution for RLHF/post-training systems. The empirical validation is highly compelling. DistFlow demonstrates substantial throughput (tokens/s) improvements over the established baseline (`verl`) across 7B, 32B, and 72B model scales. A standout achievement consistently highlighted by the reviewers is the framework's robustness in long-context scenarios: DistFlow successfully managed a 72B model at a 32k context length, whereas the centralized baseline system suffered a critical Out-Of-Memory (OOM) failure under the same conditions.
* **Rebuttal Impact:** The authors effectively addressed the minor concerns raised during the review period. Questions regarding the framework's implementation complexity, fault tolerance, and compatibility with various RL algorithms were sufficiently clarified. Furthermore, the authors provided adequate justifications and additional details regarding the asynchronous execution flows, satisfying the reviewers' requests for deeper technical clarity.

### **Conclusion**
The paper presents a robust, technically sound, and highly scalable systems solution to a critical bottleneck in modern LLM post-training. The ability to prevent OOM errors and maintain high throughput in data-intensive, long-context scenarios proves the architectural superiority of the proposed framework. The authors' rebuttal comprehensively resolved remaining reviewer questions. The submission is a clear accept.